# Oxidative Stress-Generating Antimicrobials, a Novel Strategy to Overcome Antibacterial Resistance

**DOI:** 10.3390/antiox9050361

**Published:** 2020-04-26

**Authors:** Álvaro Mourenza, José A. Gil, Luís M. Mateos, Michal Letek

**Affiliations:** Departamento de Biología Molecular, Área de Microbiología, Universidad de León, 24071 León, Spain; amouf@unileon.es (Á.M.); jagils@unileon.es (J.A.G.)

**Keywords:** oxidative stress, intracellular pathogens, antimicrobial resistance, reactive oxygen and nitrogen species

## Abstract

Antimicrobial resistance is becoming one of the most important human health issues. Accordingly, the research focused on finding new antibiotherapeutic strategies is again becoming a priority for governments and major funding bodies. The development of treatments based on the generation of oxidative stress with the aim to disrupt the redox defenses of bacterial pathogens is an important strategy that has gained interest in recent years. This approach is allowing the identification of antimicrobials with repurposing potential that could be part of combinatorial chemotherapies designed to treat infections caused by recalcitrant bacterial pathogens. In addition, there have been important advances in the identification of novel plant and bacterial secondary metabolites that may generate oxidative stress as part of their antibacterial mechanism of action. Here, we revised the current status of this emerging field, focusing in particular on novel oxidative stress-generating compounds with the potential to treat infections caused by intracellular bacterial pathogens.

## 1. Introduction

Oxidative stress is a concept that was coined by Dr. Helmut Sies in 1985 as an essential process for living beings that is based on an imbalance between oxidants and antioxidants [1]. The concept has now evolved to encompass signaling processes [2], and because of that the term oxidative stress is being replaced by redox biology. This field of knowledge can be divided into two major subfields: eustress, which is the physiological oxidative stress with metabolic purposes that is essential for redox signaling, and distress, which is considered an excess production of oxidants that may cause cellular damage [3]. In human cells, the resulting cellular damage may lead to an accumulation of errors that increase the risk to develop neurological disorders such as Parkinson´s or Alzheimer´s diseases [4], chronic metabolic illnesses such as diabetes [5] or cystic fibrosis [6], and different types of cancer [7]. However, the cellular damage produced by oxidative stress can also be used to control infections caused by bacterial pathogens. Accordingly, our innate immune cells synthesize different reactive oxygen and nitrogen species (RONS) as part of their antibacterial activity. These compounds disturb the bacterial growth and replication by different processes that are not yet fully understood [8]. To cast some light on these processes, we will summarize here the importance of RONS-generation during phagocytosis of bacterial pathogens, as well as the main bacterial mechanisms implicated on the counteraction of oxidative stress exerted by intracellular bacterial pathogens.

## 2. Oxidative Stress Response in Intracellular Bacterial Pathogens

Bacteria are exposed to different RONS synthesized by immune cells during phagocytosis [9]. However, some pathogens are able to circumvent these oxidative conditions and colonize the intracellular environment [8,10,11]. Therefore, it is very likely that the molecular pathways that maintain redox homeostasis in these bacteria are used to counteract oxidative stress during the colonization of human cells. Thanks to these mechanisms of protection to oxidative stress, intracellular bacterial pathogens might cause a high range of diseases with high morbidity and mortality in humans [12,13]. In addition, it is becoming clear that many pathogens that were considered purely extracellular, such as *Staphylococcus aureus*, are in fact able to survive facultatively within human cells during infection [14]. Moreover, antimicrobial resistance is allowing the appearance of new emerging and re-emerging bacterial pathogens that can infect host cells, making them a global concern in human health [13]. As a consequence, the mechanisms of redox homeostasis in bacteria are becoming a very attractive target for the development of new anti-infectives, and this is a very promising strategy to circumvent antimicrobial resistance.

## 3. Molecular Pathways of RONS-Biosynthesis in Immune Cells

During the innate immune response, professional phagocytes engulf bacteria when they are recognized by means of different membrane receptors, or by the action of opsonins such as immunoglobulin G (IgG) [15]. There are many different surface proteins expressed by immune cells to identify pathogens that are called generically pattern recognition receptors (PRRs), which recognize pathogen-associated molecular patterns (PAMPs) [12,15,16]. The main PRRs are integrins, toll-like receptors (TLRs), Fc receptors, the tumor necrosis factor receptor superfamily (TNFRSF), and G protein-coupled receptors (GPCRs; Figure 1A) [16].

The interaction between PRRs and PAMPs trigger the activation of the oxidative burst during phagocytosis, which is initially characterized by the activity of NADPH oxidases (NOX) that generate the superoxide anion (O_2_^−^). This free radical could be dismuted to hydrogen peroxide (H_2_O_2_) spontaneously, but superoxide dismutases do this more efficiently (Figure 1B) [8,10,17]. The presence of NOX proteins is critical for the control of infections. Accordingly, chronic granulomatous disease in humans is caused by mutations in the genes encoding NOX proteins and increases exponentially the susceptibility to recurrent bacterial and fungal infections [16,18].

The H_2_O_2_ produced during phagocytosis is able to permeate across bacterial membranes and interacts with ferrous iron (Fe^2+^) and thiol groups (-SH) of protein cysteines, which may inactivate enzymes essential for the pathogen (Figure 2) [19]. Fe^2+^ is oxidized during a Fenton reaction by H_2_O_2_ and generates the hydroxyl radical, which causes further damage to bacterial proteins, DNA, and lipids [10,20].

In addition, hypochlorous acid (HClO) is generated from H_2_O_2_ and the chloride ion (Cl^−^) by the action of myeloperoxidases (Figure 1B), which are mainly expressed in macrophages and neutrophils [10,11,17,21]. HClO has a higher antibacterial activity than H_2_O_2_ and it is more reactive with the sulfur contained in cysteines and methionines of essential proteins for the intracellular survival of the pathogen [9,22,23].

The inducible nitric oxide synthases (iNOS) are activated in later stages of phagocytosis. These enzymes produce nitric oxide (NO^•^) from l-arginine. Nitric oxide can react with the superoxide ion synthesized by NOX proteins to produce peroxynitrite (ONOO^−^; Figure 1B). Peroxynitrite can directly oxidize thiol groups of sulfur-containing amino acids; furthermore, it can be broken down to nitrogen dioxide and hydroxyl radical, which may also actively react with sulfur containing amino acid residues of bacterial proteins [11,24].

## 4. Antioxidant Systems of Intracellular Bacterial Pathogens

Intracellular bacterial pathogens are well equipped to circumvent and/or counteract the effect of the RONS produced during phagocytosis, which allows them to survive the oxidative burst and colonize the host cell. They combat oxidative stress via a complex battery of enzymatic activities that can be classified into two main groups: (i) preventative mechanisms, mainly based on protein scavengers that are able to degrade RONS; and (ii) reparation mechanisms, whose main role is the reduction of oxidized protein thiol groups to restore the activity of essential enzymes for the pathogen. These two mechanisms are clearly related since the main target of reparation enzymes are protein scavengers of RONS that are oxidized during its catalytic activity.

Both mechanisms are based on thiol switches and they are activated by transcriptional regulators that are able to interact with RONS at very high rates [10,11,24]. The most studied redox regulator is OxyR, which may act as a transcriptional activator or repressor across many different bacterial species [25,26,27]. Its extremely high constant rate with H_2_O_2_ (10^5^ M^−1^ s^−1^) makes OxyR an important oxidative stress regulator in bacteria [25,26,27,28,29].

However, there are other transcriptional regulators in bacteria that may detect small concentrations of ROS to trigger a quick response against oxidative stress [30]. The most studied are the MarR-family homologs, which are present in the genomes of many different intracellular pathogens [31]. In addition, new families of thiol-based transcriptional regulators have been recently discovered [32], and some of these are only responsive to specific RONS such as the sodium hypochlorite sensor HypS [33].

### 4.1. Enzymatic Preventative Mechanisms of Oxidative Stress

The thiol groups of the sulfur-containing thiol-based transcriptional regulators are oxidized by RONS during the early stages of phagocytosis (Figure 2). This may lead to the formation of disulphide bonds with other thiol groups of the same protein (i.e., intramolecular disulphide bond), or with thiol groups of another protein (i.e., intermolecular disulphide bond). The disulphide bond formation leads to conformational changes in the transcriptional regulators that may activate the expression of different enzymatic scavengers of RONS, such as catalases (Kat), glutathione peroxidases (GPx), or peroxiredoxins (Prx) [11,25,30,32,34].

The most studied enzymatic scavengers are superoxide dismutases (SODs). These enzymes catalyze the dismutation of O_2_^•−^ to H_2_O_2_, which is then quickly converted to H_2_O and O_2_ by catalases (Figure 3A). Both superoxide dismutases and catalases are considered important virulence factors of many intracellular pathogens [35,36,37,38,39].

On the other hand, some thiol groups of specific proteins may react relatively slowly with H_2_O_2_. These proteins, which may react with H_2_O_2_ at a highest constant rate, are called H_2_O_2_-scavengers. Thiol peroxidases were the first discovered H_2_O_2_-scavengers, and they can transduce the oxidative signal to regulate the expression of different transcriptional factors [40,41]. The thiol groups of these proteins can react with H_2_O_2_ at constant rates of 10^4^ to 10^8^ M^−1^ s^−1^.

There are two families of thiol peroxidases: peroxiredoxins (Prx) and glutathione peroxidases (GPx). Peroxiredoxins are thiol peroxidases with a well conserved structure and they usually function through a dithiol mechanism. Their enzymatic activity is controlled by two cysteines, the peroxidative cysteine (C_P_) and the resolutive cysteine (C_R_), with sequential roles during the catalytic activity [42,43]. C_P_ triggers the nucleophilic attack of H_2_O_2_, with its subsequent thiol oxidation that leads to a protein conformational change. However, C_P_ overoxidation is prevented by C_R_, which protects C_P_ by means of a disulphide bond formation (Figure 2) [43].

Glutathione peroxidases are classified into two main groups: cysteine glutathione peroxidases and selenocysteine glutathione peroxidases. However, only cysteine glutathione peroxidases (CysGPx) are present in bacteria and they show a constant reaction rate with H_2_O_2_ of 10^4^–10^5^ M^−1^s^−1^ [44]. Their catalytic activity is similar to the one described above for peroxiredoxins, i.e., two cysteines are also involved in the formation of a disulphide bond. Usually, a conformational change of the protein is triggered by the oxidation of the C_P_, which is followed by a disulphide bond formation with the thiol group of the C_R_ [45]. The importance of CysGPx during infections caused by intracellular pathogens is still understudied. However, it has been recently found that a CysGPx named GpoA is an important virulence factor of *Streptococcus pyogenes* [46].

Although thiol peroxidases are important as H_2_O_2_ sensors, catalases are considered the most important protein scavengers. Catalases are important virulence factors of many intracellular pathogens, such as *Mycobacterium tuberculosis* or *Rhodococcus equi* [35,36,38], and they may also act as peroxinitrite scavengers during redox stress [47].

### 4.2. Enzymatic Reparation Mechanisms of Protein Oxidation

The proteins involved in the reduction of enzymes that have been oxidized by RONS are essential for the survival to phagocytosis. Their initial targets are protein scavengers of RONS and their transcriptional regulators are part of the preventative mechanisms of bacteria and they are usually oxidized by RONS during the early stages of phagocytosis. Therefore, the reparation mechanisms of intracellular pathogens are considered as their second line of defense against the oxidative burst [48,49]. Moreover, these reparation mechanisms are also involved in the reduction of housekeeping proteins and other virulence factors essential for the pathogen during infection that may become oxidized during phagocytosis (Figure 3A).

The reparation mechanisms can be classified in two groups: (i) the thioredoxin/thioredoxin reductases (Trx/TrxR) and (ii) the low molecular weight-thiols (LMW-thiols)/redoxins. However, Trx/TrxR is the most common reparation mechanism and it is widely distributed in nature [48]. This redox system was discovered in 1964 by Dr. Peter Reichard´s group [50]. Since then, the number of identified proteins that are repaired by this system during oxidative stress has increased exponentially.

In bacteria, the deletion of one or more of the genes encoding thioredoxins directly alters the H_2_O_2_ resistance of the resulting mutant strain [51,52]. However, the deletion of the genes encoding thioredoxins is in many occasions not viable because of the importance of these proteins on bacterial metabolism [53,54].

In addition, it has been recently discovered a new Trx-based system made of proteins that are located on the bacterial surface, i.e., the extracellular thioredoxins (Etrx). Etrx proteins have been discovered in different pathogenic and non-pathogenic bacteria, including *M. tuberculosis*, *R. equi*, *Streptococcus pneumoniae*, *Neisseria gonorrhoeae*, *Agrobacterium tumefaciens*, and *Bradyrhizobium japonicum* [55,56,57,58,59,60,61]. The targets of the Etrx proteins are still unclear, but the deletion of the genes encoding Etrx’s abolishes the virulence of *M. tuberculosis* [55], *R. equi* [56], and *S. pneumoniae* [57,58].

On the other hand, the response to oxidative stress in many bacteria is also dependent on the protection of thiol groups of protein cysteines exerted by LMW-thiols (Figure 3B). Three different LMW-thiols have been described in bacteria, and all of them are coupled to specific redoxins. The most studied LMW-thiol is glutathione, which is coupled to glutaredoxins (GSH/Grx) and it is present in the majority of living organisms studied [49]. However, GSH/Grx is replaced by the mycothiol/mycoredoxins system (MSH/Mrx) in Actinomycetes [62], and by bacillithiol and bacilliredoxins (BSH/Brx) in Firmicutes [63]. LMW-thiols can react actively with RONS and oxidized proteins, therefore any disruption of the LMW-thiol synthesis genes affects the virulence and RONS resistance of many intracellular bacterial pathogens [64].

The redoxins coupled to LMW-thiols are also important in maintaining the redox homeostasis of many different organisms. E.g., glutaredoxins have been deeply studied in human RONS signaling [65]. The function of mycoredoxins and bacilliredoxins have been studied in different Actinomycetes and Firmicutes to understand their role in maintaining redox homeostasis under oxidative stress [62,64,66,67,68,69,70,71,72]. In addition, three recent reports have casted some light on the importance of Grx [73], Mrx [70] and Brx [67] proteins during host cell infection caused by different bacteria. However, it is becoming clear that a significant redundancy of the genes encoding these redoxins and their partially overlapping functions may complicate the analysis of their precise role in the pathogenesis of intracellular bacterial pathogens [74]. For example, we have recently demonstrated that the intracellular pathogen *R. equi* carries genes encoding three mycoredoxins with overlapping roles during host cell infection, being necessary at least one of them for intracellular survival [70]. This is important, because their partially overlapping roles may explain why other authors had not observed any attenuation in mutant strains carrying deletions on just one of the mycoredoxins present in the genome of other actinobacterial pathogens, such as *M. tuberculosis* [75]. Overall, the structure of the Brx and Mrx redoxins is well conserved among different bacterial species [67,70,76], which may explain their overlapping roles in maintaining the redox homeostasis of different intracellular pathogens.

In summary, the redox mechanisms based on thioredoxins and LMW-thiols and their reductases are essential for RONS resistance and the intracellular survival of many bacterial pathogens [64,67,77,78]. However, further research is required to understand their precise role during infection.

## 5. RONS-Producing Anti-infectives Are an Attractive Strategy to Overcome Antimicrobial Resistance

It is estimated that infections produced by antimicrobial resistant bacteria cause approximately 30,000 deaths per year in either the European Union or the United States of America. In addition, this has an associated economic burden of €1.5 billion in the EU and $20 billion in the US every year [79].

The selective pressure exerted by the abuse and misuse of antimicrobials have resulted in the selection of novel bacterial strains that are resistant to the majority of antibiotherapies currently available [80,81]. In addition, it has been estimated that two-thirds of the antibiotics used in the clinic exhibit a poor cellular uptake into eukaryotic cells. Therefore, many antimicrobials are totally ineffective against intracellular bacterial pathogens, despite that these antibiotics may have a clear bacteriostatic or bactericidal effect in vitro [82]. Because of that, new treatments against intracellular pathogens are urgently needed to solve the antimicrobial resistance crisis.

However, the development of novel anti-infectives has become unappealing to the pharma industry. This is mainly due to the fact that the development period of these drugs is much longer than their validity period, since antimicrobial resistant strains are being isolated shortly after any new antibiotic gets approval to be clinically used [81].

Because of this, drug repositioning of RONS-generating antimicrobials has gained interest in recent years (Figure 4). In particular, there have been several strategies developed to block the antioxidant systems of bacterial pathogens [83]. For instance, the antioxidant compound Ebselen (also called PZ 51, DR3305, and SPI-1005; Figure 4B), is a synthetic organoselenium-based drug with anti-inflammatory, antioxidant, and cytoprotective activities [84,85]. Ebselen may have applications in the treatment of cardiovascular disease, arthritis, stroke, atherosclerosis, and cancer, by acting as a mimic of glutathione peroxidase in mammalian cells [86,87]. However, Ebselen is also a potent inhibitor of TrxR in bacteria lacking glutathione, such as *M. tuberculosis* or *S. aureus* [88,89], which results in oxidative stress [85,90,91]. Importantly, Ebselen could also be combined with ROS-stimulating compounds that block the antioxidant defenses of bacteria such as silver nanoparticles (Figure 4) [91].

On the other hand, the use of the antimicrobial coating AGXX^®^ (Largetec GmbH, Berlin) could be a promising RONS-inspired preventative strategy against antimicrobial resistant bacteria. AGXX^®^ is made of two transition metals (silver and ruthenium), which generate oxidative stress and loss of iron homeostasis in methicillin-resistant *S. aureus* [92].

Metal oxide nanoparticles (MO-NPs), such as zinc oxide, gold or silver nanoparticles [93,94,95], are another very promising RONS-producing antibiotherapeutic strategy that could be used in combination with other RONS-generating compounds (Figure 4A). Despite of the fact that there are numerous studies demonstrating the antibacterial role of metal oxide nanoparticles [93,96], their mechanism of action based on RONS-production is still not fully understood. In many occasions, their RONS-based antimicrobial activity is activated by light. For example, titanium dioxide and zinc oxide nanoparticles are RONS-producing antimicrobials active against *S. aureus* and *Staphylococcus epiderdimis* when they are activated with blue light (at 415 nm) [97]. Similarly, other metal oxide nanoparticles (e.g., V_2_O_5_, CeO_2_, Fe_2_O_3_, and Al_2_O_3_-NPs) are O_2_^•−^ generators activated by light with potent antimicrobial activities against Gram-negative bacteria [98]. Because of this particular mechanism of activation, these metal oxide nanoparticles could only be used as topical antimicrobials due to the low penetration of visible light through the skin [99,100]. However, the production of light-activated antibacterial surfaces with polymers made with some of these nanoparticles has gained interest in recent years [101]. Other applications include the use of photoactivated TiO_2_ coatings on prostheses to prevent surgical site infections [102].

Interestingly, some traditional antibiotics can also produce RONS as part of their mechanism of action [103,104,105,106]. These antibiotics may alter the pathogen’s central metabolism and/or its iron homeostasis, which results in the production of intracellular hydrogen peroxide [107]. Recent studies have identified RONS-producing-antibiotics by expressing redox biosensors in several bacterial species (Table 1) [107,108,109,110]. In addition, the combination of different RONS-generating antimicrobials may act synergistically against some bacterial pathogens (Figure 4A) [109]. Similarly, the combination of some antibiotics with silver nanoparticles may enhance RONS biosynthesis and, therefore, increase the efficacy of the combinatorial treatment. These novel therapeutic strategies can improve the antimicrobial activity of some drugs with repurposing potential. It is even possible that antimicrobials clinically approved to treat common infections might be used as part of combinatorial therapies against new multi-drug resistant (MDR) bacteria [90,109]. However, most of this evidence is still only based on in vitro experiments, and therefore further research is required to demonstrate the efficacy of these novel treatments against pathogenic bacteria in vivo.

On the other hand, quinones (Figure 4B) are compounds that can produce thiol-depletion in many prokaryotic organisms. Their oxidative effect is derived from a one-electron reduction pathway carried by an NAD(P)H-dependent reductase that results in a semi-quinone radical formation [111]. During the incomplete reduction of quinones, the semiquinone radical resulting from this reaction may lead to the partial reduction of O_2_ to O_2_^•−^, which is a highly reactive oxygen species [112].

Quinones are compounds that can be produced as secondary metabolites by *Actinoallomorus* and *Streptomyces* sp. [113,114] and show antimicrobial activity against important pathogens such as *Enterococcus* sp., *Streptococcus* sp., *Staphylococcus* sp., or *Moraxela catarrhalis* [113]. Most importantly, quinones are active against methicillin-resistant *S. aureus* [114], which is a facultative intracellular pathogen [115].

Finally, there are several plant-derived compounds that show a clear antimicrobial activity because of their capacity to generate an oxidative shift in the bacterial cytoplasm. The most studied is allicin (Figure 4B), which is a defense molecule produced by garlic (*Allium sativum*) with important antibacterial activities and responsible of the aroma of fresh garlic [116,117]. Allicin is a reactive sulfur species (RSS), which is able to oxidize thiol groups of proteins in a dose-dependent manner. The antimicrobial activity of allicin and its oxidative role has been clearly demonstrated in *S. aureus* and *Bacillus subtilis*. In these bacteria, allicin generates a strong disulfide stress that significantly reduces the bacterial viability [118,119].

There are many other secondary metabolites produced by plants that may elicit oxidative stress in bacteria, such as catechins, ferulic acid, and their derivatives [120,121]. The combination of RONS-generating antimicrobials with these compounds may lead to the development of promising therapeutic strategies against different intracellular bacterial pathogens [120,121].

However, one major drawback of some RONS-producing antibiotics is the generation of oxidative stress on specific host tissues, which may render clinically ineffective the therapeutic strategies based on these drugs. Indeed, some aminoglycosides, fluoroquinolones, and beta-lactam antibiotics may induce host cellular damage in specific tissues such as the renal cortex or tendons by generating oxidative stress [122,123,124,125]. Nevertheless, this side effect could be lessened by specific antioxidant molecules [124,126,127].

## 6. Concluding Remarks

The maintenance of redox homeostasis is a key process that is tightly controlled by intracellular pathogens during the colonization of the host cell. This mechanism is based on different redoxins and low molecular weight-thiol molecules that maintain the bacterial cytoplasm reduced. The capacity of intracellular pathogens to respond to the oxidative stress generated by macrophages as well as their ability to circumvent or resist antimicrobials makes them an important human health issue. Novel therapeutic strategies based on the capacity of different compounds to increase RONS synthesis during phagocytosis are being developed with the aim to unbalance bacterial redox defenses and stop host cell colonization. Despite of the fact that the majority of these novel treatments have not been yet tested in vivo, they have a great potential to solve the increasing problem of antibiotic-resistant infections caused by intracellular bacterial pathogens.

## Figures and Tables

**Figure 1 antioxidants-09-00361-f001:**
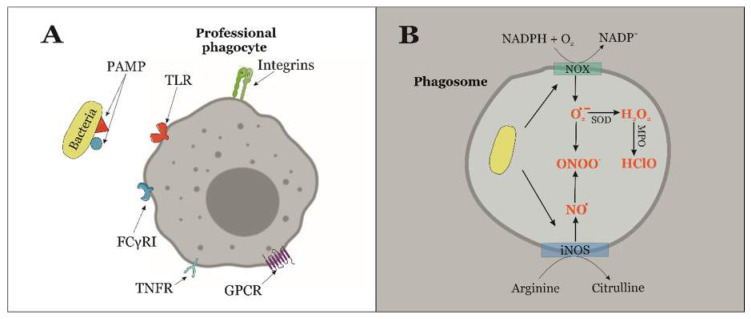
(**A**) During infection, pathogen-associated molecular patterns (PAMPs) are recognized by pattern recognition receptors (PRRs) which are present on the surface of professional phagocytes. PRRs include integrins, toll-Like receptors (TLRs), Fc receptors such as FCγRI, tumor necrosis factor receptors (TNFRs), and G Protein-Coupled Receptors (GPCRs). (**B**) RONS synthesis is triggered during phagocytosis to generate a bactericidal oxidative stress. There are different enzymes involved on this process, including NADPH oxidases (NOX), nitric oxide synthases (iNOS), superoxide dismutates (SOD), and myeloperoxidases (MPO).

**Figure 2 antioxidants-09-00361-f002:**
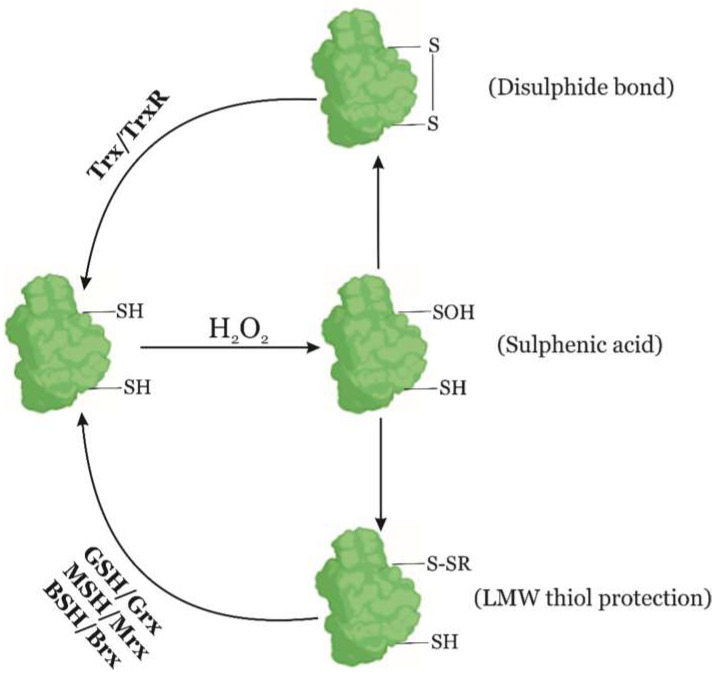
Bacterial redox mechanisms. When a cysteine of a target protein (in green) is oxidized by RONS its thiol group is converted to sulfenic acid (-SOH). To prevent overoxidation, the sulfenic acid is reduced by other thiol groups of the protein generating a disulphide bond, or by low molecular weight-thiols (LMW-thiols; S-SR). Eventually, thioredoxins (Trx) reduce the oxidized cysteine residues and break the disulphide bond, whereas LMW-thiols are reduced back by glutaredoxins (Grx), mycoredoxins (Mrx), or bacilliredoxins (Brx).

**Figure 3 antioxidants-09-00361-f003:**
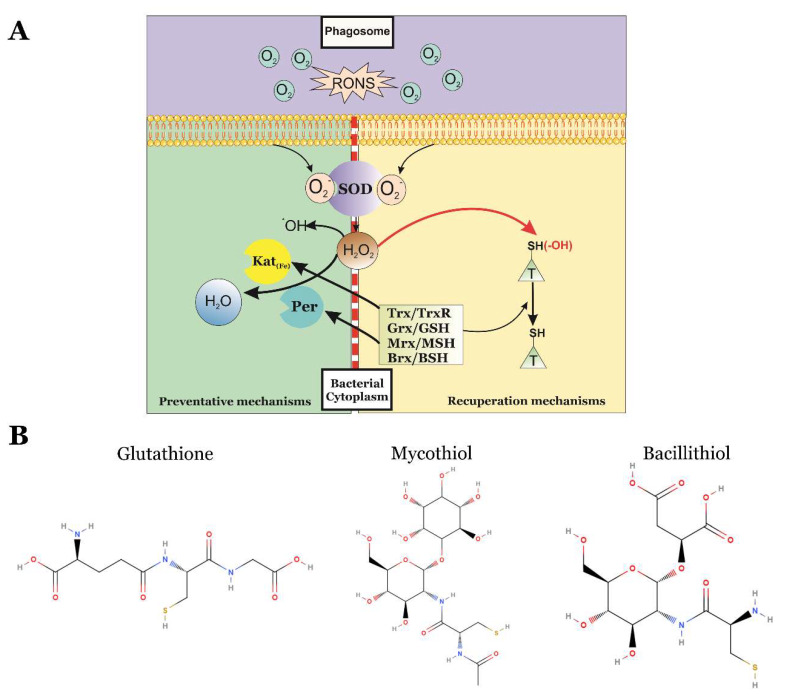
(**A**) After phagocytosis, the RONS biosynthesis produced during the oxidative burst triggers the activation of the redox mechanisms of intracellular bacterial pathogens. Their preventative mechanisms are activated to degrade RONS. If these preventative measures are not sufficient, the recuperation mechanisms will restore the reduced state of protein scavengers and other oxidized proteins. SOD, superoxide dismutases; Kat, catalases; Per, peroxidases; Trx/TrxR, thioredoxin/thioredoxin reductases; GSH/Grx, glutathione/glutaredoxins; MSH/Mrx, mycothiol/mycoredoxins; BSH/Brx, bacillithiol/bacilliredoxins; T, target proteins. (**B**) Chemical structures of low molecular weight-thiols described in bacteria.

**Figure 4 antioxidants-09-00361-f004:**
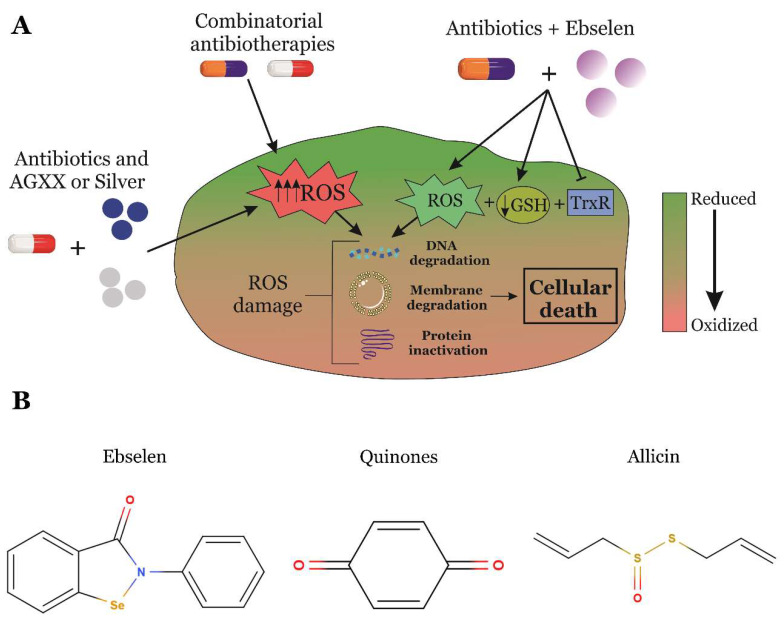
(**A**) Novel strategies based on the repositioning of antibiotics and the combination of different RONS-producing antibacterial compounds that may be used to treat infections caused by antibiotic-resistant strains. (**B**) Chemical structures of novel RONS-generating antimicrobials.

**Table 1 antioxidants-09-00361-t001:** List of RONS-generating antimicrobials, their primary mechanism of action and microorganisms on which their ability to produce oxidative stress was tested.

Antibiotic	Primary Mechanism of Action	Microorganism	Reference
Erythromycin	Protein synthesis inhibition	*Rhodococcus equi*	[109]
Rifampicin	RNA synthesis inhibition	*Rhodococcus equi*	[109]
Vancomycin	Cell wall synthesis inhibition	*Mycobacterium tuberculosis* *Rhodococcus equi* *Staphylococcus aureus*	[106] [109] [103]
Norfloxacin	DNA gyrase inhibition	*Rhodococcus equi* *Staphylococcus aureus* *Escherichia coli*	[109] [103] [104]
Clofazimine	DNA replication inhibition	*Mycobacterium tuberculosis*	[106]
Ethambutol	Cell wall synthesis inhibition	*Mycobacterium tuberculosis*	[106]
Isoniazid	Cell wall synthesis inhibition	*Mycobacterium tuberculosis*	[106]
Quinones	Different cellular targets	*Enterococcus* sp. *Streptococcus* sp. *Staphylococcus* sp. *Moraxela catarrhalis*	[113] [113] [113] [113]
Metal oxide nanoparticles	Undefined	*Escherichia coli* *Staphylococcus aureus* *Staphylococcus epiderdimis* *Photobacterium phosphoreum*	[91] [97] [97] [98]

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
