# Peer review of "Oxidative Stress-Generating Antimicrobials, a Novel Strategy to Overcome Antibacterial Resistance"

_antioxidants, 2020, doi:10.3390/antiox9050361_

Round 1

Reviewer 1 Report

Nicely written, up-to-date review with concise language. Please find my comments below. Speaking about the concept of the manuscript, I am missing the discussion on the selectivity of RONS-generating compounds. For sure they will affect not only the pathogen but also the host’s system. I would urge the authors to add a paragraph covering this issue (where authors can state some facts and express their opinion on the selectivity/usability of RONS-generating compounds).

  • Generally, abbreviations should not be used in the Title. I kindly ask the authors and the editorial office to decide whether ‘RONS’ is a widely acknowledged abbreviation to such extent that it can be used in the title (and keywords) without definition.
  • line 75 – What is meant by ‘specialized’ in ‘specialized thiol groups of protein cysteines’? Please explain or remove.
  • Caption to Figures 1 and 3 – consider explaining the abbreviations used in the figure. Now the explanations are explained only in the text (sometimes far from the figure itself, which lowers the readability)
  • In caption to Fig 2, the authors stated that thioredoxins (Trx) convert subphrenic acid to thiol, but un the figure, the Trx enzyme catalyses the reaction from disulphide to thiol. Explanation is needed.
  • line 145 – ‘orange’ – there is nothing orange in the figure
  • line 232 – ‘ebselen……acting as a mimic of glutathione peroxidase’. The concept that a small molecule of ebselen mimicking the function of peroxidase needs to be explained. Why is antioxidant ebselen, which is a compound lowering the levels of H202. discusses together with compounds, which do exactly the opposite – increase the oxidative stress. This section needs clarification for sure.
  • lines 251-253 – When talking about metal oxides nanoparticles activated by light, please add a short discussion, whether these strategies are applicable only for topical application (considering how deep the visible light – and blue light especially) can penetrate the skin).
  • Table 1 – the binomial names of microorganisms should not be abbreviated – for example R. equi is not easy to decipher from non-microbiologists.
  • As a medicinal chemist, I was missing the structures of the discussed compounds. Consider adding the structure of ebselen and general structure of quinones at least.

Author Response

Many thanks for your comments, our paper has greatly improved after we addressed all of your points. Please find below our answers.

Nicely written, up-to-date review with concise language. Please find my comments below. Speaking about the concept of the manuscript, I am missing the discussion on the selectivity of RONS-generating compounds. For sure they will affect not only the pathogen but also the host’s system. I would urge the authors to add a paragraph covering this issue (where authors can state some facts and express their opinion on the selectivity/usability of RONS-generating compounds).

We fully agree with the reviewer, we have now covered this point in the discussion (Line 306).

Generally, abbreviations should not be used in the Title. I kindly ask the authors and the editorial office to decide whether ‘RONS’ is a widely acknowledged abbreviation to such extent that it can be used in the title (and keywords) without definition.

The abbreviation has been replaced in the title and keywords with “oxidative stress” or “reactive oxygen and nitrogen species”, respectively.

line 75 – What is meant by ‘specialized’ in ‘specialized thiol groups of protein cysteines’? Please explain or remove.

Thanks for pointing this out, we have removed this word.

Caption to Figures 1 and 3 – consider explaining the abbreviations used in the figure. Now the explanations are explained only in the text (sometimes far from the figure itself, which lowers the readability)

Many thanks for this comment, we have explained all abbreviations in both figure captions.

In caption to Fig 2, the authors stated that thioredoxins (Trx) convert subphrenic acid to thiol, but un the figure, the Trx enzyme catalyses the reaction from disulphide to thiol. Explanation is needed.

Our apologies, the figure is correct, i.e. the Trx enzyme catalyses the reaction from disulphide to thiol, but the figure caption was unclear on this point. Therefore, this figure legend has been amended.

line 145 – ‘orange’ – there is nothing orange in the figure

Thanks for spotting this mistake, some parts of a previous version of the figure were orange. This has been now modified (current line 149).

line 232 – ‘ebselen……acting as a mimic of glutathione peroxidase’. The concept that a small molecule of ebselen mimicking the function of peroxidase needs to be explained. Why is antioxidant ebselen, which is a compound lowering the levels of H202. discusses together with compounds, which do exactly the opposite – increase the oxidative stress. This section needs clarification for sure.

Many thanks for this comment, this paragraph was indeed quite confusing and it has been rewritten to clarify this point. We agree with the reviewer, Ebselen acts as an antioxidant in mammalian cells. However, Ebselen is also a potent inhibitor of TrxR in bacteria lacking glutathione, and this inhibition leads to oxidative stress. We have now included this clarification in lines 241-245.

lines 251-253 – When talking about metal oxides nanoparticles activated by light, please add a short discussion, whether these strategies are applicable only for topical application (considering how deep the visible light – and blue light especially) can penetrate the skin).

Thanks, this point has been covered in lines 263-268.

Table 1 – the binomial names of microorganisms should not be abbreviated – for example R. equi is not easy to decipher from non-microbiologists.

We have removed all abbreviations of species names listed in Table 1.

As a medicinal chemist, I was missing the structures of the discussed compounds. Consider adding the structure of ebselen and general structure of quinones at least.

We have added the chemical structures of these compounds to Figure 4.

Reviewer 2 Report

This is a well structured, informative and timely review which contextualises recent developments and future opportunities for research/developments in oxidative stress inducing antimicrobial chemotherapies.

A very useful, succinct summary that will be of value to thos actively engageed in this research arena, buat also presented in a manner which will make it an informative first source of information for non-specialist researchers who are interested in exploring the topic

the paper is well written and structured. I recommend publication with only a few minor suggestions/corrections:

  1. would be useful to have a diagram showing the different molecular structures of GSH, MSH, BSH to highlight their structural differences, but functional equivalence in different bacteria.
  2.  would be useful to define some of the abbreviations used in Fig.1 and Fig.3 within the figure legends as this will make the figures much easier to follow and understand

Author Response

This is a well structured, informative and timely review which contextualises recent developments and future opportunities for research/developments in oxidative stress inducing antimicrobial chemotherapies.

A very useful, succinct summary that will be of value to thos actively engageed in this research arena, buat also presented in a manner which will make it an informative first source of information for non-specialist researchers who are interested in exploring the topic

the paper is well written and structured. I recommend publication with only a few minor suggestions/corrections:

would be useful to have a diagram showing the different molecular structures of GSH, MSH, BSH to highlight their structural differences, but functional equivalence in different bacteria.

 would be useful to define some of the abbreviations used in Fig.1 and Fig.3 within the figure legends as this will make the figures much easier to follow and understand

Many thanks for your kind comments. We have added the molecular structures of GSH, MSH and BSH to Figure 3. In addition, we have defined all abbreviations used in Fig. 1 and Fig. 3 in their figure captions.